# CONFESS: A FRAMEWORK FOR SINGLE SOURCE CROSS-DOMAIN FEW-SHOT LEARNING

**Debasmit Das, Sungrack Yun & Fatih Porikli**
Qualcomm AI Research.*
{debadas,sungrack,fporikli}@qti.qualcomm.com

## ABSTRACT

Most current few-shot learning methods train a model from abundantly labeled base category data and then transfer and adapt the model to sparsely labeled novel category data. These methods mostly generalize well on novel categories from the same domain as the base categories but perform poorly for distant domain categories. In this paper, we propose a framework for few-shot learning coined as **ConFeSS** (**Con**trastive Learning and **Fe**ature **S**election **S**ystem) that tackles large domain shift between base and novel categories. The first step of our framework trains a feature extracting backbone with the contrastive loss on the base category data. Since the contrastive loss does not use supervision, the features can generalize better to distant target domains. For the second step, we train a masking module to select relevant features that are more suited to target domain classification. Finally, a classifier is fine-tuned along with the backbone such that the backbone produces features similar to the relevant ones. To evaluate our framework, we tested it on a recently introduced cross-domain few-shot learning benchmark. Experimental results demonstrate that our framework outperforms all meta-learning approaches and produces competitive results against recent cross-domain methods. Additional analyses are also performed to better understand our framework.

## 1 INTRODUCTION

Recently, there has been an expansion in the quality and quantity of datasets (Zhang et al., 2018; Sun et al., 2017), computing resources (Jeon *et al.*, 2019), and deep neural architectures (Dhillon & Verma, 2020). When trained with vast amounts of data, these deep neural network models deliver improved performance on applications like image recognition, action localization, speaker verification, text analysis, and gene sequence prediction (Nguyen et al., 2018; Yun et al., 2019; Yao et al., 2019; Zhou *et al.*, 2018). However, data collection and annotation at a large scale incurs substantial labor and cost, which are particularly difficult for specialized domains such as medical imaging and satellite imagery, where domain expertise is needed. Moreover, most neural networks fail to generalize to unseen categories when trained with a few labeled samples. To address these limitations, research on few-shot learning has gained significant attention.

Few-shot learning methods (Wang et al., 2020) aim to uncover the data structure and model the concept of new categories with only a few labeled samples. A popular strategy to tackle few-shot learning is meta-learning which consists of two stages: *meta-train* and *meta-test*. In the meta-train stage, a backbone network is trained to classify the base category correctly by leveraging the labeled source data while mimicking a few-shot regime where only a limited number of samples are available per class in each learning episode. In the meta-test stage, with the trained backbone, the novel categories with only a few target class samples can be added (or enrolled) and tested. Here, the backbone network can be adapted to the target samples.

Nonetheless, most few-shot learning approaches exhibit insufficient generalization capacity when there is a big gap between the source and target data. To investigate this problem, there have been considerable efforts (Triantafillou et al., 2020; Chen et al., 2019; Tseng et al., 2020) in establishing cross-domain few-shot learning (CDFSL) benchmarks. Still, these datasets limit their focus to natural

---

*Qualcomm AI Research is an initiative of Qualcomm Technologies, Inc.

images and fail to capture more pragmatic domain shifts where target data may come from more diverse domains such as satellite and medical imagery. Very recently, Guo et al. (2020) introduced a challenging benchmark to evaluate generalization capability on distant target domains. Its target domain datasets consist of images from natural, medical and satellite domains with wide variations of context, color, and perspective. Thus, it represents practical applications where the generic model needs to be adapted to a particular use case. On this benchmark, the popular meta-learning approaches (Vinyals et al., 2016; Finn et al., 2017; Snell et al., 2017; Sung et al., 2018; Lee et al., 2019) have been found to produce poor recognition performance.

This paper proposes a novel contrastive learning and feature selection system (ConFeSS) for single-source cross-domain few-shot learning. Our framework consists of three steps: pre-training a backbone network on a single-source dataset, learning a feature masking module on the target dataset, and fine-tuning the backbone network. In the first step, a backbone network is trained in an unsupervised fashion, where a self-supervised learning approach is considered with the contrastive loss (Chen et al., 2020). This is in contrast to meta-learning approaches, which use supervision during the pre-training stage. Although the label of the source dataset is given at this step, we consider the unsupervised learning to alleviate the *supervision collapse* problem (Doersch et al., 2020) and also to generalize better to the distant target domains. In the second step, a feature masking module is learned with target domain data to generate masks for separating task-relevant features from irrelevant features. This step is required because there is a large discrepancy between the source and target datasets, and hence all the features useful for the source task might not be helpful or even be detrimental to the target task. Furthermore, we expect the generalization performance to be improved with fewer features in the few-shot regime due to the Vapnik-Chervonenkis (VC) dimension reduction. In the final step, both the pre-trained backbone network and the classifier are fine-tuned to adapt to the target categories by a proper regularization with the relevant features.

Our main contributions can be summarized as follows: (i) Learning a feature masking module with appropriate constraints to select relevant features for few-shot target samples; (ii) Fine-tuning the backbone by regularizing it with the selected relevant features; (iii) Our extensive experimental evaluation and analyses show that our method produces competitive recognition performance on the new CDFSL benchmark (Guo et al., 2020).

## 2 RELATED WORK

**Meta-learning for Few-shot Learning** These methods use episodic pre-training to simulate test conditions followed by fast adaptation to novel category samples. One of the earliest meta-learning methods was MatchNet (Vinyals et al., 2016) which learns a mapping function to project labeled and unlabeled samples to their corresponding labels. ProtoNet (Snell et al., 2017) extended this work by learning a representation and assigning a class depending on the distance of query samples to class prototypes while RelationNet (Sung et al., 2018) learns an additional deep metric function. MetaOpt (Lee et al., 2019) takes a different approach where an SVM-like classifier is learned on top of the features for better generalization. Finally, MAML (Finn et al., 2017) is an optimization-based method that learns to adapt to few-shot novel categories in a few iterations. All these meta-learning methods have performed poorly on the CDFSL benchmark (Guo et al., 2020). There are many other meta-learning works but they have not been evaluated on the CDFSL benchmark. One can refer (Hospedales et al., 2021) for a comprehensive survey on this topic.

**Domain Adaptation** In this problem, we have source and target domains with the same categories, and the goal is to reduce domain discrepancy between them. Hence, domain adaptation methods cannot be directly used for CDFSL, where the labels between source and target are disjoint. The universal domain adaptation (UDA) setting (You et al., 2019) might be more similar to the CDFSL setting because it has different source and target categories. However, in UDA, there is some overlap between the source and the unknown target categories with lots of unlabeled target data available while CDFSL considers completely novel target categories each containing only few labeled data.

**Cross-domain Few-shot Learning** There have been very few works on cross-domain few-shot learning. A recent work (Tseng et al., 2020) uses a noisy transformation layer on top of features to simulate cross-domain distributions and produce better generalization. In (Chen et al., 2019), the authors compare different meta-learning frameworks and propose a competitive fine-tuning-based baseline against these methods for the cross-domain setting. However, the datasets used for evaluating

these methods contain only natural images. As a result, there is no significant domain shift between the source and target datasets even though the source and target labels are disjoint. Guo et al. (2020) introduce a novel CDFSL benchmark and show that most meta-learning methods along with the feature-wise transformation (Tseng et al., 2020) approach perform poorly compared to simple fine-tuning methods. In our paper, the fine-tuning step is augmented with a feature selection mechanism to select relevant features. More recent methods that evaluate on the CDFSL benchmark include CHEF (Adler et al., 2020), ATA (Wang & Deng, 2021) and STARTUP (Phoo & Hariharan, 2021). CHEF addresses large domain shift by fusion of Hebbian learners applied on different layers. This is done to increase the importance of low and mid-level features for distant domain recognition. ATA is a plug-and-play method that improves robustness of models through adversarial task augmentation. STARTUP assumes access to large unlabelled data from the target domain and proposed combining knowledge distillation and contrastive learning to learn the target model. In our framework, we only assume access to few labeled data from the target domain.

**Self-supervision for Few-shot Learning** Self-supervised learning has been used in the form of different pretext tasks (He et al., 2020; Noroozi & Favaro, 2016; Gidaris et al., 2018) to pre-train representations that can be used for down-stream tasks as well. These representations have been able to generalize well in the few-shot regime. Recent works (Gidaris et al., 2019; Su et al., 2020; Chen et al., 2021) show that adding self-supervised loss functions for representation learning improves few-shot recognition performance. In our paper, we solely use self-supervision in the form of contrastive loss (Chen et al., 2020) during pre-training because it mitigates supervision collapse as observed in (Doersch et al., 2020). Furthermore, contrastive losses have been theoretically proven (Saunshi et al., 2019) to produce better representations for few-shot learning but have not been evaluated for their generalization ability on few-shot novel categories with distant domains.

**Feature Selection for Few-shot Learning** Feature selection is useful for deriving relevant features for a particular task or for preventing overfitting on few-shot samples. Zhao et al. (2018) separate the features into orthogonal components where the sparse signal component facilitates the feature selection. It is similar to our approach where we use a mask to select relevant and irrelevant features, yet we impose different constraints on these decomposed features. Liu et al. (2017) use a greedy feature selection mechanism followed by multiple dropouts to reduce gradient variance of few-shot samples. However, this method is not applicable for transferring to novel tasks with large domain differences. A more recent work (Dvornik et al., 2020) select features from a universal representation learned from multiple source domains by optimizing the selection coefficients for different domains. This is quite different from our method, which can work even with a single source domain by selecting relevant features instead of relevant source domains. Berriel et al. (2019) use budget-aware mechanism of optimizing a switch vector to select domain-relevant feature channels from a pre-trained architecture. Additionally, masking has been used to adapt single network weights to multiple new tasks (Mancini et al., 2018; Mallya et al., 2018).

# 3 PROPOSED FRAMEWORK

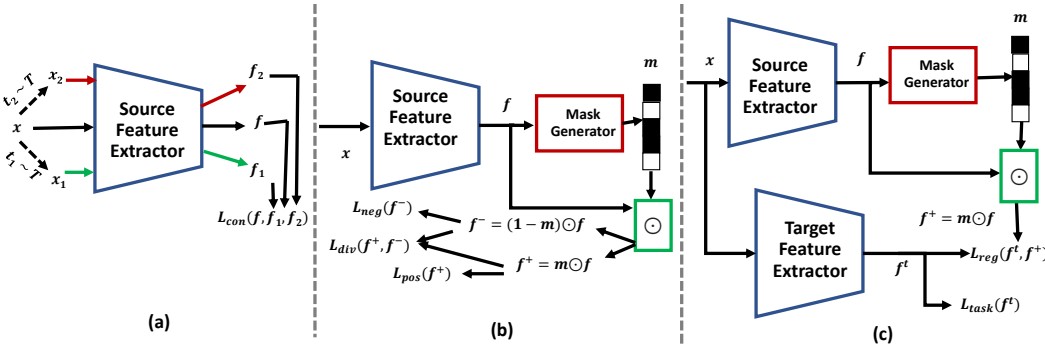

Figure 1: Our framework consists of three steps: (a) Pre-training the backbone using a self-supervised contrastive loss; (b) Learning the masking module on the target data to select relevant features, and (c) Fine-tuning the backbone using a regularized loss with positively relevant features.

### 3.1 PROBLEM DESCRIPTON AND NOTATION

For the CDFSL problem, we have a source domain and a target domain. Each domain has an associated joint distribution $P$ over the input space $\mathcal{X}$ and the label space $\mathcal{Y}$. The marginal distribution of the input space is denoted as $P_{\mathcal{X}}$. Instances $(x, y)$ can be sampled from $P$, where $x$ is the input and $y$ is the corresponding label. Accordingly, the source domain can be represented as $(\mathcal{X}_s, \mathcal{Y}_s)$ and the target domain as $(\mathcal{X}_t, \mathcal{Y}_t)$ with joint distributions $P_s$ and $P_t$, respectively. Due to the domain difference, the source marginal distribution $P_{\mathcal{X}_s}$ will be significantly different from the target marginal distribution $P_{\mathcal{X}_t}$. Moreover, the target domain classes are novel; hence there is no overlap between $\mathcal{Y}_s$ and $\mathcal{Y}_t$. The goal is to first learn a model from abundant data sampled from the source distribution $P_s$. Then the model is adapted to few data sampled from the target distribution $P_t$. Finally, the adapted model is evaluated on held-out test data sampled from the target distribution. In our framework, we learn the model from the source distribution, using a self-supervised contrastive loss function. The adaptation step on the target data involves learning a mask generator followed by regularized fine-tuning. Our framework is depicted in Fig. 1, and the details are described in the following subsections.

### 3.2 UNSUPERVISED TRAINING OF BACKBONE

The backbone (the feature extraction module) is trained in an unsupervised manner inspired from recent works on contrastive learning (Chen et al., 2020) and unsupervised pre-training (Doersch et al., 2020). Contrastive learning has been found to be effective for transfer learning (**?**). This is because contrastively learned features focus more on mid and low-level features, which are easily adapted. Furthermore, such features produce better reconstruction by learning a holistic representation of images rather than focusing only on discriminative regions. Thus, contrastive learning is an effective pre-training strategy for transferring representations to distant target domains. In our pre-training stage, we augment samples from the existing samples in the training batch using various transformations and use these augmented samples and original samples to determine a contrastive loss. Specifically, let there be $N_b$ training samples in a batch, where the samples are represented as $\{\mathbf{x}_i\}_{i=1}^{N_b}$. For each sample $\mathbf{x}_i$, we obtain $N_t$ random transformations where the $t^{th}$ transformed instance is represented as $\mathbf{x}_{it}$ and $t \in \{1, 2..., N_t\}$. Following the idea of (Doersch et al., 2020), we enforce the the transformed instances $\mathbf{x}_{it}$ to be close to $\mathbf{x}_i$ and far from $\mathbf{x}_k$, $k \neq i$ using the following cross-entropy loss,

$$L_{con} = -\frac{1}{N_b N_t} \sum_{i=1}^{N_b} \sum_{t=1}^{N_t} \log \frac{\exp(-d(\phi_s(\mathbf{x}_{it}), \phi_s(\mathbf{x}_i)))}{\sum_{k=1}^{N_b} \exp(-d(\phi_s(\mathbf{x}_{it}), \phi_s(\mathbf{x}_k)))}. \tag{1}$$

Here, $\phi_s(\cdot)$ represents the feature extraction module, and $d(\cdot)$ is a distance metric. Snell et al. (2017) showed that Euclidean distances model Bregman divergence of mixture densities, which consistently performs better for the few-shot setting, and so we choose the same metric. The appendix discusses the theoretical support of constrastive learning for few-shot learning.

### 3.3 LEARNING THE FEATURE MASKING MODULE

The feature masking module is used to generate masks that can select task-relevant and task-irrelevant features. For simplicity, we call task-relevant and irrelevant features positive and negative features, respectively. It is important to note that we cannot afford a large masking sub-network because it might overfit to few-shot target domain samples. So, we just mask on features fed to the classifier with appropriate regularization during fine-tuning. Let the feature extraction module learned from the source domain be denoted as $\phi_s(\cdot)$. Given a batch of target domain samples $\{(\mathbf{x}_i, y_i)\}_{i=1}^N$, we can obtain the feature $\mathbf{f}_i = \phi_s(\mathbf{x}_i) \in \mathbb{R}^d$ for each sample. We feed the feature into the mask generating module $M(\cdot)$ to obtain the mask $\mathbf{m}_i = M(\mathbf{f}_i)$. This mask is then used to produce positive ($\mathbf{f}_i^+$) and negative ($\mathbf{f}_i^-$) features, such that

$$\mathbf{f}_i^+ = \mathbf{m}_i \odot \mathbf{f}_i, \qquad \mathbf{f}_i^- = (\mathbf{1} - \mathbf{m}_i) \odot \mathbf{f}_i \tag{2}$$

where $\odot$ is the Hadamard product, and $\mathbf{1}$ is a vector of ones of the appropriate dimension. $\mathbf{m}_i \in \mathbb{R}^d$ is a mask vector consisting of $d$ elements where the $j^{th}$ element is represented as $m_{ij}$. To generate binary masks $m_{ij}$, we follow the probabilistic procedure introduced in (Maddison et al., 2017; Jang et al., 2017). Let $z_{ij}$ be the unbounded output logit from the mask module corresponding to the

$i^{th}$ sample and the $j^{th}$ dimension. We generate logistic noise $l$ such that $l = \log(u) - \log(1 - u)$ and $u \sim \text{uniform}(0, 1)$. The noise is then added to the logits to produce mask $m_{ij}$, such that $m_{ij} = \sigma(\frac{z_{ij}+l}{\tau})$ where $\sigma(\cdot)$ is the sigmoid operation, and $\tau$ is the temperature scale. The noise is added to the logits to explore different binary masks suitable for the target task. To back-propagate discrete masks during training, we follow the straight-through estimator (Bengio et al., 2013) where we use sigmoid during the backward pass and hard-threshold operation during the forward pass. The hard-threshold operation involves setting $m_{ij}$ to 1 if $m_{ij} > 0.5$ or 0 otherwise. During inference mode, the hard-threshold operation of the mask is carried out but with the logistic noise $l = 0$

To train the feature masking module $M(\cdot)$, we want to make sure that the positive features $\mathbf{f}_i^+$ are discriminative while the negative features $\mathbf{f}_i^-$ are not. To produce discriminative positive features $\mathbf{f}_i^+$, we use the cross-entropy criterion such that

$$L_{pos}(\mathbf{f}_i^+) = L_{XEnt}(C^+(\mathbf{f}_i^+), y_i), \tag{3}$$

where $L_{XEnt}(\cdot)$ is the cross-entropy criterion, and $C^+(\cdot)$ is a linear classifier used for the positive features $\mathbf{f}_i^+$. To produce negative features $\mathbf{f}_i^-$, we use the maximum entropy criterion such that

$$L_{neg}(\mathbf{f}_i^-) = -L_{Ent}(C^-(\mathbf{f}_i^-)), \tag{4}$$

where $L_{Ent}(\cdot)$ is the entropy of the softmax outputs of $C^-(\mathbf{f}_i^-)$, and $C^-(\cdot)$ is a linear classifier used for the negative features $\mathbf{f}_i^-$. The maximum entropy criterion makes sure that output class probabilities are uncertain causing the negative features to be less class discriminative.

The design of the mask only makes positive and negative features apart. However, they can still be statistically similar in arrangement of clusters and higher-order statistics. Hence, a divergence measure to maximize the statistical distance between the positive and the negative features is required. If we let $s_d(\cdot)$ be the statistical distance between two sets of features: the positive set $\mathbf{F}^+ = \{(\mathbf{f}_i^+)_{i=1}^N\}$ and the negative set $\mathbf{F}^- = \{(\mathbf{f}_i^-)_{i=1}^N\}$, then we would minimize the divergence loss,

$$L_{div}(\mathbf{F}^+, \mathbf{F}^-) = e^{-s_d(\mathbf{F}^+ . \mathbf{F}^-)}. \tag{5}$$

The exponent term is used to provide more stable and smaller gradients when close to optimality. The loss terms in Eq. 3, 4 and 5 are weighted and combined to obtain

$$L_{mask} = \lambda_{pos}L_{pos} + \lambda_{neg}L_{neg} + \lambda_{div}L_{div}. \tag{6}$$

Here, $L_{pos}$ and $L_{neg}$ are averaged over the batch samples, while $L_{div}$ is an aggregated loss function over all the batch samples. These loss terms are combined to obtain the final loss $L_{mask}$, which is back-propagated across $M(\cdot)$, $C^+(\cdot)$ and $C^-(\cdot)$ to update the respective parameters.

### 3.4 FINE-TUNING

The fine-tuning stage is the final step of adaptation to the target domain. In this step, we train both the feature extractor and the classifier on the target domain data. Since the target domain contains only a few labeled data, we regularize the feature extractor to produce positive features using the mask generator that has been trained in the previous step. Let $\phi_t(\cdot)$ be the target domain feature extractor that is initialized from the parameters of the source domain feature extractor $\phi_s(\cdot)$. Given a batch of target domain samples $\{(\mathbf{x}_i, y_i)\}_{i=1}^N$, for each sample we can obtain the feature $\mathbf{f}_i^t = \phi_t(\mathbf{x}_i) \in \mathbb{R}^d$. This feature $\mathbf{f}_i^t$ is fed into a linear classifier $C(\cdot)$ such that we obtain the cross-entropy loss,

$$L_{task}(\mathbf{f}_i^t) = L_{XEnt}(C(\mathbf{f}_i^t), y_i). \tag{7}$$

To regularize the network, we want to make sure that the target domain feature $\mathbf{f}_i^t$ is close to the target relevant (positive) feature $\mathbf{f}_i^+ = M(\phi_s(\mathbf{x}_i)) \odot \phi_s(\mathbf{x}_i)$. This is realized by minimizing the loss,

$$L_{reg} = ||\mathbf{f}_i^t - \mathbf{f}_i^+||_2^2, \tag{8}$$

where $||\cdot||_2$ is the 2-norm. The regularization ensures that the network does not catastrophically forget the positively relevant features and does not allow the negatively relevant features to be transferred. Additionally, distance-based regularization has been shown to promote tighter generalization (Gouk et al., 2021) as discussed in the appendix. The loss terms in Eq. 7 and 8 are combined as

$$L_{ft} = L_{task} + \lambda_{reg}L_{reg}. \tag{9}$$

$L_{ft}$ is then averaged over the training samples in a batch to compute the final loss, which is back-propagated across $\phi_t(\cdot)$ and $C(\cdot)$ to update the respective parameters. We can choose not to fine-tune $\phi_t(\cdot)$ but ablation studies in Table 2 show that fine-tuning backbone is effective for CDFSL. This completes the fine-tuning stage. All the stages of our proposed framework, including the pre-training and the fine-tuning steps, are summarized in **Algorithm 1**. For a test sample $\mathbf{x}_{te}$, we use $C(\phi_t(\mathbf{x}_{te}))$ followed by the softmax operation to obtain the class probabilities and the most probable class.

---

**Algorithm 1:** ConFeSS framework

---

**Given:** Source dataset $\mathcal{D}_s$ and Target dataset $\mathcal{D}_t$
**Hyper-parameters:** $\lambda_{pos}, \lambda_{neg}, \lambda_{div}, \lambda_{reg}$
**Step 1:** Pre-train backbone $\phi_s(\cdot)$ on $\mathcal{D}_s$
**For** each sampled batch of source data
   **For** each sampled augmentation
     Take gradient descent step of Eq. 1 with respect to $\phi_s(\cdot)$
**Step 2:** Obtain mask generator $M(\cdot)$ from $\mathcal{D}_t$
**For** each sampled batch of target data
   Take gradient descent step of Eq. 6 with respect to $M(\cdot), C^+(\cdot)$ and $C^-(\cdot)$
**Step 3:** Fine-tune backbone $\phi_t(\cdot)$ on $\mathcal{D}_t$
Initialize $\phi_t(\cdot)$ from optimized $\phi_s(\cdot)$
**For** each sampled batch of target data
   Take gradient descent step of Eq. 9 with respect to $\phi_t(\cdot)$ and $C(\cdot)$
**Step 4:** Predict test sample class using optimized $\phi_t(\cdot)$ and $C(\cdot)$

---

## 4 EXPERIMENTAL RESULTS

### 4.1 DATASET DESCRIPTION

To evaluate our proposed framework, we test it on the CDFSL benchmark introduced by Guo et al. (2020). This benchmark uses mini-ImageNet (Vinyals et al., 2016), which is a subset of the Im-ageNet (Deng et al., 2009) dataset as the source domain that contains abundantly labeled natural categories. The model learned on the mini-Imagenet dataset is then tested on target datasets containing only a few labeled training data. These target datasets have large domain differences from the source domain, and in order of increasing dissimilarity, they consist of the following: a) CropDiseases (Mohanty et al., 2016), containing images of different plant disease types, b) EuroSAT (Helber et al., 2019), consisting of different classes of satellite imagery, c) ISIC2018 (Tschandl *et al.*, 2018; Codella et al., 2018), which contains different dermoscopic images of skin lesions, and d) ChestX (Wang *et al.*, 2017), a collection of chest X-Ray images of different lung disease types.

Table 1: Results of our approach (ConFeSS) as compared with previous methods on the **ChestX**, **ISIC**, **EuroSAT** and **CropDisease** datasets. The best results are shown in boldface. NWKS means N-way K-shot test setting.

| Method | ChestX | | | ISIC | | | EuroSAT | | | CropDisease | | |
|---|---|---|---|---|---|---|---|---|---|---|---|---|
| | 5W5S | 5W20S | 5W50S | 5W5S | 5W20S | 5W50S | 5W5S | 5W20S | 5W50S | 5W5S | 5W20S | 5W50S |
| MatchNet | 22.40 | 23.61 | 22.12 | 36.74 | 45.72 | 54.58 | 64.45 | 77.10 | 54.44 | 66.39 | 76.38 | 58.53 |
| MatchNet+FWT | 21.26 | 23.23 | 23.01 | 30.40 | 32.01 | 33.17 | 56.04 | 63.38 | 62.75 | 62.74 | 74.90 | 75.68 |
| MAML | 23.48 | 27.53 | – | 40.13 | 52.36 | – | 71.70 | 81.95 | – | 78.05 | 89.75 | – |
| ProtoNet | 24.05 | 28.21 | 29.32 | 39.57 | 49.50 | 51.99 | 73.29 | 82.27 | 80.48 | 79.72 | 88.15 | 90.81 |
| ProtoNet+FWT | 23.77 | 26.87 | 30.12 | 38.87 | 43.78 | 49.84 | 67.34 | 75.74 | 78.64 | 72.72 | 85.82 | 87.17 |
| RelationNet | 22.96 | 26.63 | 28.45 | 39.41 | 41.77 | 49.32 | 61.31 | 74.43 | 74.91 | 68.99 | 80.45 | 85.08 |
| RelationNet+FWT | 22.74 | 26.75 | 27.56 | 35.54 | 43.31 | 46.38 | 61.16 | 69.40 | 73.84 | 64.91 | 78.43 | 81.14 |
| MetaOpt | 22.53 | 25.53 | 29.35 | 36.28 | 49.42 | 54.80 | 64.44 | 79.19 | 83.62 | 68.41 | 82.89 | 91.76 |
| STARTUP | 26.94 | 33.19 | 36.91 | 47.22 | 58.63 | 64.16 | 82.29 | 89.26 | 91.99 | **93.02** | **97.51** | **98.45** |
| CHEF | 24.72 | 29.71 | 31.25 | 41.26 | 54.30 | 60.86 | 74.15 | 83.31 | 86.55 | 86.87 | 94.78 | 96.77 |
| FT-All | 25.97 | 31.32 | 35.49 | 48.11 | 59.31 | **66.48** | 79.08 | 87.64 | 90.89 | 89.25 | 95.51 | 97.68 |
| ATA | 24.43 | – | – | 45.83 | – | – | 83.75 | – | – | 90.59 | – | – |
| ConFeSS | **27.09** | **33.57** | **39.02** | **48.85** | **60.10** | 65.34 | **84.65** | **90.40** | **92.66** | 88.88 | 95.34 | 97.56 |

### 4.2 IMPLEMENTATION DETAILS

For a fair comparison, we use the ResNet-10 backbone introduced by Guo et al. (2020), which produces a 512 dimension feature space. We use Adam as the optimizer with a learning rate of 0.001. The statistical distance $s_d(\cdot)$ used in Eq. 5 is maximum mean discrepancy (MMD) (Gretton et al., 2012). MMD between two distributions $P$ and $Q$ over feature space $\mathcal{X}$ is defined as $||\mathbb{E}_{X \sim P}[\phi(X)] -$

$\mathbb{E}_{Y \sim Q}[\phi(Y)]||_{\mathcal{H}}$ where $\phi : \mathcal{X} \to \mathcal{H}$, and $\mathcal{H}$ is a reproducing kernel Hilbert space. The MMD can be easily computed using the kernel trick while we use the Gaussian kernel in our experiments. Unless explicitly mentioned, we use the pre-training batch size $N_b = 50$ and the augmentation size $N_t = 3$, where the augmentations were chosen as in (Chen et al., 2020) . For larger values of $N_t$, we found a dip in performance probably because the extra augmentations do not represent transformations in the target domain. The results of different values of $N_t$ are reported and analyzed later in the paper. The masking module $M(\cdot)$ consists of a small two-layer feed-forward network with a hidden layer dimension of 256. We use a small subnetwork for masking to prevent overfitting issues. We set temperature $\tau = 1$. Also, we set $\lambda_{pos} = 10^{-3}$, $\lambda_{neg} = 10^{-2}$, $\lambda_{div} = 10^{-2}$, and $\lambda_{reg} = 10^{-2}$. The numbers of training epochs for Step 1, Step 2, and Step 3 in **Algorithm 1** are set as 400, 15, and 50, respectively. The hyper-parameters are kept fixed because it is not possible to create a small-to-medium validation set from the few-shot target dataset. The epoch number of 400 for pre-training is kept the same as fine-tuning methods described in (Guo et al., 2020). For all experiments, the average accuracy over 600 episodes of N-way K-shot setting is reported. Each episode contains randomly sampled K-shot samples per class for adaptation and 15 query samples per class for evaluation, where N is the number of sampled classes.

## 4.3 COMPARISONS

We compare our proposed approach against several meta-learning based few-shot learning methods introduced in the CDFSL benchmark (Guo et al., 2020): MatchNet (Vinyals et al., 2016), MAML (Finn et al., 2017), ProtoNet (Snell et al., 2017), RelationNet (Sung et al., 2018), and MetaOpt (Lee et al., 2019). Furthermore, Feature-wise Transformation (FWT) (Tseng et al., 2020) was added to the backbones to simulate the cross-domain setting of the benchmark. We also include the FT-All (Guo et al., 2020) baseline for the comparison that fine-tunes the full network with only the cross-entropy loss. Additionally, we compare with recent methods, STARTUP, CHEF and ATA, which have been evaluated on the CDFSL benchmark. It is to be noted that SENet (Hu et al., 2017) scales channels similar to the way we select features. However, it does not decompose features into relevant and irrelevant ones. Besides, SENet cannot tackle domain-shift because it is only used as a module in a feature extraction block. Hence, using SENet as the backbone for comparison would not be useful (or even fair) because all the compared baselines use the ResNet-10 backbone. The results of the comparison for 5-way 5-shot, 5-way 20-shot, and 5-way 50-shot test settings are shown in Table 1.

From Table 1, we see that our proposed framework ConFeSS outperforms all meta-learning methods by a large margin. Specifically, for the 5-shot setting, our method produces improvements of 12.64 %, 21.72 %, 15.50 % and 11.49 % over the best meta-learning method on the ChestX, ISIC, EuroSAT, and CropDisease datasets, respectively. Veritably, the improvement margin increases further as the number of shots increases, especially for medical datasets such as ChestX and ISIC. This is because medical datasets have similar classes and require more annotations to perform reasonably well. Also, the FWT module fails to generalize to target datasets and sometimes negatively affects these methods. As expected, the performance of our framework also follows the rank of domain similarity with miniImagenet: least performance for ChestX and best performance for CropDisease.

The meta-learning methods use supervision for pre-training and cannot mimic distant domain datasets, which causes them to overfit source data with poor generalization to distant target domains. In comparison, our contrastively learned backbone only learns the inherent structure of data transformations and can generalize more effectively to other domains. Secondly, the masking module selects only relevant features for fine-tuning on the target domain, thus preventing overfitting. Our method also performs better than CHEF on all settings. Compared to FT-All, ATA and STARTUP, our method achieves much higher scores in most settings except for the CropDisease dataset, which is the easiest benchmark for the cross-domain task as it is the most similar to miniImageNet. STARTUP uses large amount of unlabelled target domain data while our proposed approach does not use any. Still, our method outperforms STARTUP in more difficult 9 out of 12 settings.

## 4.4 ADDITIONAL ANALYSES

**Ablation study:** Table 2 shows ablation study results. The masking module uses losses functions $L_{neg}$ and $L_{div}$. Also, the fine-tuning step uses loss $L_{reg}$. w/o Feature Mask implies that the pre-trained network is fine-tuned using only cross-entropy loss, without using a masking module. We can also choose not to fine-tune the whole backbone, which is denoted as w/o FT BB in Table 2 for

Table 2: Ablation results. ↑ shows increase in performance with ablation.

| Setting | 5-way 5-shot | | | | 5-way 20-shot | | | |
|---|---|---|---|---|---|---|---|---|
| | CropDisease | EuroSAT | ISIC | ChestX | CropDisease | EuroSAT | ISIC | ChestX |
| Full Framework | 88.88 | 84.65 | 48.85 | 27.09 | 95.34 | 90.40 | 60.10 | 33.57 |
| w/o Cont. Learn. | 87.26 | 83.15 | 47.66 | 26.06 | 95.47 ↑ | 88.78 | 59.96 | 32.12 |
| w/o FT BB | 85.18 | 83.14 | 42.25 | 25.76 | 93.52 | 89.70 | 52.61 | 31.12 |
| w/o Feature Mask | 87.57 | 83.87 | 47.10 | 26.09 | 95.49 ↑ | 89.93 | 61.08 ↑ | 33.20 |
| w/o $L_{div}$ | 87.95 | 84.23 | 48.62 | 26.92 | 94.74 | 89.31 | 59.20 | 32.77 |
| w/o $L_{neg}$ | 89.03 ↑ | 84.41 | 48.04 | 26.60 | 94.72 | 90.37 | 60.14 ↑ | 32.81 |
| w/o $L_{reg}$ | 87.83 | 83.98 | 48.34 | 26.73 | 94.43 | 90.31 | 59.83 | 32.69 |
| Direct Positive | 87.15 | 83.94 | 47.25 | 26.58 | 93.65 | 89.40 | 59.66 | 31.92 |

the 5-way 5-shot and 5-way 20-shot settings. As shown, in most cases, removing these components result in a drop in performance, suggesting that all these loss functions and components are essential. An important step in the framework is fine-tuning the whole backbone, where the most significant drop is observed when it is absent. This indicates that for large domain differences between source and target domain, fine-tuning the backbone is essential. In the 5-shot setting, there is always a drop in performance when removing the feature masking module. However, for the 20-shot setting, performance improves slightly on the ISIC and CropDisease datasets. This demonstrates that the feature masking module is more critical for fewer shot settings. Among $L_{div}$, $L_{reg}$, and $L_{neg}$, there is no clear winner since their order of importance depends on the dataset and the shot. In the table, w/o Cont. Learn. implies when the contrastive pre-training step is replaced by traditional supervised pre-training using cross-entropy loss. The results show that using supervised pre-training produces lower recognition accuracy than contrastive pre-training. However, the standard supervised pre-training along with masking and fine-tuning still performs better than most of the other compared methods in Table 1. We also consider the Direct Positive setting for the ablation study, where instead of using $L_{reg}$, we directly use the positive features obtained using the mask generator to fine-tune the feature extractor and classifier. The results obtained using this technique are competitive compared to previous work but still worse than the Full framework and w/o $L_{reg}$ ablation.

**The number of features selected:** We also analyzed the number of features selected for different datasets in Fig. 2 (a). According to VC theory, the number of features selected for 5-shot setting is less than that of the 20-shot setting to prevent overfitting. The number of features selected for the 50-shot setting is less because the 50-shot setting has lots of training samples and does not require a masking module. Also, CropDisease has the highest number of features selected while ChestX has the least number of features selected because CropDisease dataset contains natural images and is more similar to miniImageNet.

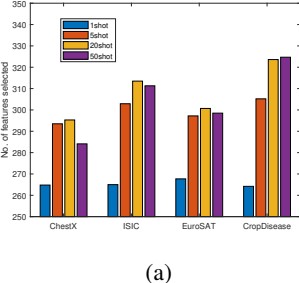 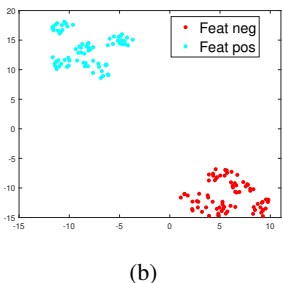 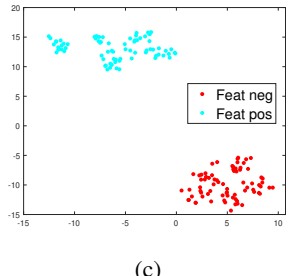

(a)          (b)          (c)

Figure 2: (a) Average number of positive features selected for different datasets and shots. T-SNE plot of positive and negative features of CropDisease dataset for (b) 5-way 5-shot, (c) 5-way 20-shot setting

**Comparison between positive and negative features:** We also investigate the difference between the positive and negative features. In practice, the positive features are more relevant to the classification task, and therefore expected to be more discriminative than the negative features. We quantify the discrimination ability of features using the metric $DS = Tr(S_b)/Tr(S_w)$, where $S_b$ is the between-class scatter matrix, $S_w$ is the within-class scatter matrix and $Tr(\cdot)$ is the trace operation. The scatter matrices are defined as $S_b = \sum_{i=1}^{N} n_i(\mu_i - \mu)(\mu_i - \mu)^T$ and $S_w = \sum_{j=1}^{M}(x_j - \mu_{y_j})(x_j - \mu_{y_j})^T$, where $\mu_i$ is the sample mean of the $i^{th}$ class, $\mu$ is the mean of all the samples, and $n_i$ is the number of samples in the $i^{th}$ class with a total of $N$ classes. $(x_j, y_j)$ is the $j^{th}$ sample-label pair out of $M$ total samples. Higher values of $DS$ indicate better discrimination. We compared $DS$ across different shots and datasets in Table 3 for the 5-way setting. As expected, $DS$ scores for the positive features are higher than that of the negative features for both the 5-shot and 20-shot settings. The $DS$ scores get higher for the positive features of the 20-shot setting since more training samples produce better

Table 3: Discrimination scores (DS) of positive (+) and negative features (-) for different shots (S) and datasets

| Setting | ChestX | | | | ISIC | | | | EuroSAT | | | | CropDisease | | | |
|---|---|---|---|---|---|---|---|---|---|---|---|---|---|---|---|---|
| | 5S+ | 5S- | 20S+ | 20S- | 5S+ | 5S- | 20S+ | 20S- | 5S+ | 5S- | 20S+ | 20S- | 5S+ | 5S- | 20S+ | 20S- |
| DS | 0.018 | 0.016 | 0.03 | 0.02 | 0.13 | 0.10 | 0.17 | 0.09 | 0.67 | 0.42 | 0.62 | 0.28 | 0.63 | 0.41 | 0.77 | 0.31 |

clusters. The $DS$ difference between positive and negative features for the ChestX dataset is low mostly because the dataset is very hard to cluster. Visualization using t-SNE (Maaten & Hinton, 2008) for 5-shot and 20-shot settings are shown in Fig. 2 (b) and (c) respectively. Results show that positive features produce better clusters compared to negative ones. As expected, the positive features for the 20-shot setting are more discriminative compared to the 5-shot setting. Also, the internal statistics of positive and negative features are different because of $L_{div}$, which maximizes the statistical distance between the two sets of features.

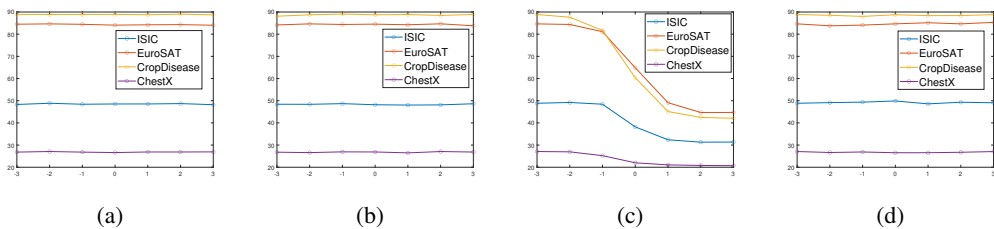

|   (a)   |   (b)   |   (c)   |   (d)   |

Figure 3: Accuracy with 5W5S setting as (a) $\log_{10} \lambda_{neg}$ (b) $\log_{10} \lambda_{div}$ , (c) $\log_{10} \lambda_{reg}$ and (d) $\log_{10} \lambda_{pos}$ vary.

**Hyper-parameter sensitivity:** We study our framework's performance as the hyper-parameters $\lambda_{neg}$, $\lambda_{pos}$, $\lambda_{div}$, and $\lambda_{reg}$ are varied. The results for the 5-shot setting are shown in Fig. 3, which shows that the recognition performance is stable with respect to $\lambda_{neg}$, $\lambda_{pos}$, and $\lambda_{div}$, while the performance drops for larger values of $\lambda_{reg}$. This is because $\lambda_{neg}$, $\lambda_{pos}$, and $\lambda_{div}$ affect the learning of a much smaller feature masking network. On the other hand, the value of $\lambda_{reg}$ affects the learning of a much larger network - the target feature extractor, which eventually plays a direct role in the inference stage. The plot in Fig. 3 (c) shows that we should choose $\lambda_{reg} < 1$ for better performance.

**The impact of the number of augmentations:** We report how the number of augmentations $N_t$ used in contrastive pre-training affects cross-domain few-shot recognition performance. Results for 5-way 5-shot, 5-way 20-shot, and 5-way 50 shot settings are shown in Table 4. Results show a sharp drop in

Table 4: Recognition performance on the N-way K-shot setting as $N_t$ is varied during pre-training.

| Dataset/$N_t$ | 5-way 5-shot | | | | 5-way 20-shot | | | | 5-way 50-shot | | | |
|---|---|---|---|---|---|---|---|---|---|---|---|---|
| | 3 | 10 | 20 | 30 | 3 | 10 | 20 | 30 | 3 | 10 | 20 | 30 |
| CropDisease | 88.88 | 70.11 | 72.28 | 70.51 | 95.34 | 86.21 | 86.85 | 87.08 | 97.56 | 92.48 | 92.10 | 92.42 |
| EuroSAT | 84.65 | 62.28 | 60.38 | 59.93 | 90.40 | 72.45 | 70.40 | 69.50 | 92.66 | 76.87 | 75.99 | 75.29 |
| ISIC | 48.85 | 37.59 | 38.08 | 36.64 | 60.10 | 48.60 | 48.80 | 47.99 | 65.34 | 53.86 | 54.78 | 53.78 |
| ChestX | 27.09 | 23.30 | 23.29 | 23.16 | 33.57 | 25.68 | 25.24 | 25.37 | 39.02 | 27.72 | 27.34 | 27.25 |

recognition performance when the number of augmentations is increased beyond 3. This is because additional augmentations in the source dataset do not represent the possible augmentations in the target datasets. With higher $N_t$, there is a propensity to have augmentations that are not valid for target classes. The target datasets consist of specialized domains like medical and satellite imagery, which also exuberate inconsistent categories when the target datasets are transformed using arbitrary source augmentation policies. As a result, contrastive representations learned using those augmentations might not generalize well. For example, in ChestX, random cropping or Gaussian blur might affect discriminative regions in images. This phenomenon has also been recently studied in (Xiao et al., 2021), where color augmented representations do not transfer well for color discrimination tasks.

## 5 CONCLUSION

We presented a framework called ConFeSS (Contrastive Learning and Feature Selection System) to learn a generalizable representation followed by a feature selection mechanism while fine-tuning on the target domain. We introduce novel loss constraints on selecting relevant and irrelevant features for the target domain. Extensive experiments conducted on the cross-domain few-shot learning benchmark show our approach's advantages over the meta-learning and other CDFSL methods. Additional analyses also provide insights into the feature selection mechanism and justify the importance of each component of our framework.

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

## A    Additional Experimental Details

All our experiments were conducted in a cluster of nodes where the nodes contain NVIDIA Tesla V100 GPUs with a mix of 16GB and 32GB memory. The details of the benchmark used for comparison have been introduced in the following repository: https://github.com/IBM/cdfsl-benchmark. We only consider the single source domain setting where the source domain is miniImageNet [1], and the target domains are ChestX [2], ISIC [3], EuroSAT [4] and CropDisease [5]. Additional experimental details include: (a) Image size: $224 \times 224$ (b) Batch size during adaptation: 5 (c) MMD kernel numbers: 5 (d) MMD kernel multiplier: 2.0. The masking module architecture is as follows: [Linear(512,256) - ReLU - Linear(256,512) - Gumbel Sigmoid]

## B    Implementation of MMD

We use a multi-kernel approach to implement MMD. Specifically, for the positive and negative feature matrix: $\mathbf{F}^+ \in \mathbb{R}^{N \times d}$ and $\mathbf{F}^- \in \mathbb{R}^{N \times d}$, $MMD = \text{mean}(XX + YY - XY - YX)$. Here, $[XX]_{ij} = \sum_{k=0}^{n_k-1} \exp(-\frac{||\mathbf{F}_{i:}^+ - \mathbf{F}_{j:}^+||_2^2}{bm^k})$, $[YY]_{ij} = \sum_{k=0}^{n_k-1} \exp(-\frac{||\mathbf{F}_{i:}^- - \mathbf{F}_{j:}^-||_2^2}{bm^k})$, $[XY]_{ij} = \sum_{k=0}^{n_k-1} \exp(-\frac{||\mathbf{F}_{i:}^+ - \mathbf{F}_{j:}^-||_2^2}{bm^k})$ and $[YX]_{ij} = \sum_{k=0}^{n_k-1} \exp(-\frac{||\mathbf{F}_{i:}^- - \mathbf{F}_{j:}^+||_2^2}{bm^k})$. $\mathbf{F}_{i:}^+$ is the $i^{th}$ row of $\mathbf{F}^+$.

$m$ is the kernel multiplier and $n_k$ is the number of kernels. $b$ is the bandwidth and is computed as $b = \frac{\text{sum}(D)}{(4N^2 - 2N)(m^{\text{floor}(0.5n_k)})}$ where $[D]_{ij} = ||\mathbf{F}_{i:} - \mathbf{F}_{j:}||_2^2$ and $\mathbf{F} \in \mathbb{R}^{2N \times d}$ is the concatenation of $\mathbf{F}^+$ and $\mathbf{F}^-$.

## C    Data Augmentation for Contrastive Pre-training

The augmentation policies are the same as those used for pre-training the SimCLR framework (Chen et al., 2020). They include the following transformations:

- **Random Crop and Resize:** The random cropping has scale in the range $[0.08, 1.0]$ and aspect ratio in the range $[3/4, 4/3]$. The random cropping is always followed by a horizontal flip with each flip type having a probability of 0.5. This is followed by resizing of image to the desired size.

- **Color Distortion:** Color jitter is applied with a probability of 0.8. The jitter strength values of brightness, contrast, saturation and hue are set as 0.8, 0.8, 0.8 and 0.2, respectively. This is followed by color drop operation (convert to grayscale) with a probability of 0.2.

- **Gaussian Blur:** This is applied with a probability of 0.5, and radius of blur is selected randomly from the range $[0.1, 2.0]$.

## D    Effect of Different Ways

We also test our framework on higher number of ways *i.e.* higher number of classes per episodic evaluation. The results of the experiment are shown in Table 5. As expected, higher number of ways leads to drop in performance because of more difficulty in discrimination. Surprisingly, the drop in performance is less for easier datasets like EuroSAT and CropDisease. This might be probably because the additional classes for higher ways leads to less confusion compared to those of ChestX and ISIC.

---

[1] 100 categories. Downloaded from: https://drive.google.com/file/d/1uxpnJ3Pmmwl-6779qiVJ5JpWwOGl48xt/view

[2] 7 categories. License and download information available at: https://www.kaggle.com/nih-chest-xrays/data

[3] 7 categories. License and download information available at: https://challenge.isic-archive.com/data#2018

[4] 10 categories. Downloaded from: http://madm.dfki.de/files/sentinel/EuroSAT.zip

[5] 38 categories. License and download information available at: https://www.kaggle.com/saroz014/plant-disease

Table 5: Recognition results of our approach with different ways and shots (**S**) on the **ChestX**, **ISIC**, **EuroSAT** and **CropDisease** datasets. The entries with – imply that testing is not possible because of lesser number of total categories present.

| | ChestX | | | ISIC | | | EuroSAT | | | CropDisease | | |
|---|---|---|---|---|---|---|---|---|---|---|---|---|
| Setting | 5S | 20S | 50S | 5S | 20S | 50S | 5S | 20S | 50S | 5S | 20S | 50S |
| 5-way | 27.09 | 33.57 | 39.02 | 48.85 | 60.10 | 65.34 | 84.65 | 90.40 | 92.66 | 88.88 | 95.34 | 97.56 |
| 7-way | 20.72 | 27.15 | 32.33 | 40.38 | 52.57 | 59.29 | 80.23 | 87.66 | 90.90 | 85.37 | 94.00 | 96.51 |
| 10-way | – | – | – | – | – | – | 76.03 | 84.59 | 88.15 | 82.10 | 92.63 | 95.48 |
| 19-way | – | – | – | – | – | – | – | – | – | 75.31 | 88.45 | 93.39 |

# E  COMPARISON IN 1-SHOT SETTING

Of all the compared methods, only STARTUP (Phoo & Hariharan, 2021) and ATA (Wang & Deng, 2021) have been evaluated on the 1-shot setting. Hence, we evaluate our model on the 1-shot setting and report the results in Table 6. Results show that our method is still competitive with respect to existing methods.

Table 6: Comparison on the 1-shot setting.

| | ChestX | ISIC | EuroSAT | CropDisease |
|---|---|---|---|---|
| Method | 1S | 1S | 1S | 1S |
| STARTUP | 23.09 | 32.66 | 63.88 | 75.93 |
| ATA | 22.14 | **34.70** | **65.94** | **77.82** |
| ConFeSS | **23.67** | 33.46 | 65.51 | 76.49 |

# F  COMPARISON WITH UNSUPERVISED META-TRAINING

We also compare our method with UMTRA (Khodadadeh et al., 2019) - an unsupervised meta-training framework derived from the popular meta-learning framework MAML (Finn et al., 2017). UMTRA uses a similar algorithm as MAML but extends it to the case of unlabeled training data. Class membership for unlabeled data is determined such that a sample and its augmentation belong to the same class. We compare against two versions: UMTRA-ProtoNet and UMTRA-ProtoTune as reported in (Medina et al., 2020) on the CDFSL benchmark (Guo et al., 2020). UMTRA-ProtoTune extends UMTRA-ProtoNet by fine-tuning on target domain data. The results in Table 7 show that our method outperforms the two variants of UMTRA on all shots and all datasets.

Table 7: Comparison against unsupervised meta-training with different shots (**S**) and 5-way setting on the **ChestX**, **ISIC**, **EuroSAT** and **CropDisease** datasets.

| | ChestX | | | ISIC | | | EuroSAT | | | CropDisease | | |
|---|---|---|---|---|---|---|---|---|---|---|---|---|
| Method | 5S | 20S | 50S | 5S | 20S | 50S | 5S | 20S | 50S | 5S | 20S | 50S |
| UMTRA-ProtoNet | 24.94 | 28.04 | 29.88 | 39.21 | 44.62 | 46.48 | 74.91 | 80.42 | 82.24 | 79.81 | 86.84 | 88.44 |
| UMTRA-ProtoTune | 25.00 | 30.41 | 35.63 | 38.47 | 51.60 | 60.12 | 68.11 | 81.56 | 85.05 | 82.67 | 92.04 | 95.46 |
| ConFeSS | **27.09** | **33.57** | **39.02** | **48.85** | **60.10** | **65.34** | **84.65** | **90.40** | **92.66** | **88.88** | **95.34** | **97.56** |

# G  EFFECT OF DIFFERENT EPOCH NUMBERS

The numbers of epochs for training the masking module and fine-tuning are fixed at 15 and 50 respectively because we do not have validation split from few-shot target domain dataset to set an optimal value. The epoch numbers (15 and 50) are kept substantially low compared to pre-training epoch number (400) so that the network is less prone to over-fitting on few-shot data. To show the effect of different epoch numbers, we perform the following experiments on the 5-way 5-shot setting. Firstly, the number of epochs for training the masking module is varied and then the number of epochs for fine-tuning is varied. In Table 8, we show the results for varying epochs for training masking module while keeping epochs for fine-tuning fixed at 50 as well as the results for varying epochs for fine-tuning while keeping epochs for training masking module fixed at 15. From the results, it seems that the recognition performance is not very sensitive to the number of epochs used for training masking module or for fine-tuning. However, increasing fine-tuning epochs beyond 50 tends to decrease performance slightly for ISIC, EuroSAT and CropDisease datasets.

Table 8: Effect of varying epochs for training masking module and for fine-tuning. In the first column, number without parantheses is the epoch number for training masking module while that with parantheses is the epoch number for fine-tuning. The results are shown in corresponding labelled super-columns.

| Setting | Varying epoch number for training masking module | | | | (Varying epoch number for fine-tuning) | | | |
|---|---|---|---|---|---|---|---|---|
| | ChestX | ISIC | EuroSAT | CropDisease | ChestX | ISIC | EuroSAT | CropDisease |
| 5 (25) | 26.50 | 48.82 | 83.64 | 87.64 | 26.82 | 48.54 | 84.23 | 88.27 |
| 10 (50) | 27.26 | 48.24 | 83.90 | 88.78 | 27.09 | 48.85 | 84.65 | 88.88 |
| 15 (75) | 27.09 | 48.85 | 84.65 | 88.88 | 27.09 | 48.23 | 83.10 | 88.56 |
| 20 (100) | 26.87 | 48.48 | 84.18 | 88.01 | 27.33 | 47.83 | 82.86 | 88.23 |
| 25 (125) | 27.27 | 48.34 | 83.69 | 88.04 | 27.07 | 47.48 | 82.31 | 88.05 |

## H    ALTERNATIVE MODEL DESIGN CHOICES

In this section, we consider the following alternative model designs and report recognition performance on the 5-way 5-shot, 5-way 20-shot and 5-way 50-shot setting. The evaluation setup is similar to that described in Section 4.2:

- **L1 norm:**  In this design of the ConFeSS framework, we just replace the L2 norm $||\mathbf{f}_i^t - \mathbf{f}_i^+||_2^2$ in Eq. 8 with the L1 norm $||\mathbf{f}_i^t - \mathbf{f}_i^+||_1^2$.

- **Source Mask:**  In this setup, we train the two layer mask module (defined in appendix A) on the source dataset rather than the target dataset. The source dataset used is miniImageNet. Specifically, the mask module is trained on top of the contrastively learned pre-trained feature extractor with the miniImageNet dataset. In the final step, the target feature extractor is fine-tuned on the target dataset.

- **Neg. Reg.:**  In this setup, we use negative features for the regularization. Specifically, we use $||\mathbf{f}_i^t - \mathbf{f}_i^-||_2^2$ in Eq. 8 instead of $||\mathbf{f}_i^t - \mathbf{f}_i^+||_2^2$.

- **Dir. Neg.:**  We also consider the Direct Negative setting for the ablation study, where instead of using $L_{reg}$, we directly use the negative features obtained using the mask generator to fine-tune the feature extractor and classifier.

- $K$ **Layer Mask Mod.:**  Here, we study the effect of having masking module of different sizes. Here, $K$ stands for the number of layers used for the masking module. Specifically, we study the effect for $K = 3, 4, 5$. The masking module architecture for $K = 3, 4, 5$ are [Linear(512,256) - BatchNorm1D(256) - ReLU - Linear(256,128) - BatchNorm1D(128) - ReLU - Linear(128, 512) - Gumbel Sigmoid], [Linear(512,256) - BatchNorm1D(256) - ReLU - Linear(256,128) - BatchNorm1D(128) - ReLU - Linear(128, 256) - BatchNorm1D(256) - ReLU - Linear(256, 512) - Gumbel Sigmoid], and [Linear(512,256) - BatchNorm1D(256) - ReLU - Linear(256,128) - BatchNorm1D(128) - ReLU - Linear(128, 64) - BatchNorm1D(64) - ReLU - Linear(64, 128) - BatchNorm1D(128) - ReLU - Linear(128, 512) - Gumbel Sigmoid], respectively.

- **Joint Training:**  We consider the setup where the the masking module and the target feature extractor are trained together in one stage instead of the proposed two stages, using combined losses in Eq. 6 and Eq. 9.

The results of comparing these alternative model designs with our proposed framework ConFeSS are shown in Table 9. Results show that among all these alternative model designs, especially L1 norm, Source Mask, Neg. Reg., and Dir. Neg. perform poorly compared to our original ConFeSS framework. In most of the cases, having a larger masking module produces similar or slightly better performance compared to ConFeSS because of better representation capacity of output masks. Joint training of masking module and target feature extractor produces poorer recognition performance for 5 shot setting compared to ConFeSS. However, for higher shot setting, the joint training procedure produces similar or better performance compared to ConFeSS. This is because joint training encompasses optimization of larger number of parameters, which might cause the network to overfit on lower shots while exploit additional amount of training data for higher shots.

## I    RESULTS WITH CONFIDENCE INTERVAL

In this section, we re-report comparison studies: Table 10 and Table 11 show the performance with 95 % confidence interval for Table 1 and Table 2, respectively.

Table 9: Recognition performance on alternative model designs along with 95 % confidence interval shown in parentheses for different shots and datasets.

| Method | ChestX | | | ISIC | | | EuroSAT | | | CropDisease | | |
|---|---|---|---|---|---|---|---|---|---|---|---|---|
| | 5W5S | 5W20S | 5W50S | 5W5S | 5W20S | 5W50S | 5W5S | 5W20S | 5W50S | 5W5S | 5W20S | 5W50S |
| L1 norm | 25.78 (0.44) | 31.79 (0.47) | 38.41 (0.52) | 45.45 (0.35) | 56.40 (0.35) | 62.32 (0.30) | 81.29 (0.42) | 89.81 (0.39) | 90.34 (0.17) | 83.80 (0.28) | 92.65 (0.67) | 96.19 (0.23) |
| Source Mask | 25.82 (0.34) | 30.59 (0.47) | 36.55 (0.21) | 44.04 (0.32) | 53.69 (0.24) | 61.17 (0.43) | 79.57 (0.60) | 85.33 (0.48) | 90.78 (0.39) | 85.12 (0.38) | 90.26 (0.25) | 94.75 (0.24) |
| Neg. Reg. | 24.05 (0.11) | 28.21 (0.25) | 29.32 (0.23) | 39.57 (0.28) | 49.50 (0.51) | 51.99 (0.50) | 73.29 (0.37) | 82.27 (0.62) | 85.48 (0.35) | 79.72 (0.46) | 88.15 (0.33) | 90.81 (0.22) |
| Dir. Neg. | 24.23 (0.41) | 27.48 (0.36) | 30.32 (0.22) | 39.17 (0.44) | 48.24 (0.38) | 50.72 (0.51) | 72.69 (0.36) | 81.34 (0.55) | 83.98 (0.17) | 80.32 (0.41) | 86.15 (0.12) | 91.24 (0.53) |
| 3 Layer Mask Mod. | 26.79 (0.36) | 34.12 (0.28) | 40.04 (0.12) | 48.84 (0.27) | 60.26 (0.23) | 65.88 (0.46) | 84.13 (0.34) | 90.59 (0.23) | 91.52 (0.27) | 88.36 (0.37) | 95.72 (0.41) | 97.67 (0.10) |
| 4 Layer Mask Mod. | 27.19 (0.38) | 34.11 (0.22) | 40.34 (0.20) | 48.79 (0.18) | 60.79 (0.29) | 65.95 (0.11) | 84.16 (0.32) | 90.63 (0.26) | 91.70 (0.28) | 88.38 (0.15) | 95.56 (0.44) | 97.61 (0.33) |
| 5 Layer Mask Mod. | 26.96 (0.42) | 34.63 (0.21) | 39.79 (0.20) | 48.70 (0.37) | 60.24 (0.15) | 66.39 (0.39) | 83.74 (0.23) | 90.65 (0.14) | 91.20 (0.14) | 88.10 (0.50) | 95.93 (0.56) | 97.64 (0.12) |
| Joint Training | 25.93 (0.41) | 34.33 (0.32) | 39.57 (0.30) | 47.24 (0.41) | 59.62 (0.23) | 66.12 (0.34) | 83.01 (0.30) | 90.23 (0.12) | 91.50 (0.38) | 88.27 (0.43) | 95.71 (0.24) | 97.60 (0.18) |
| ConFeSS | 27.09 (0.24) | 33.57 (0.31) | 39.02 (0.12) | 48.85 (0.29) | 60.10 (0.33) | 65.34 (0.45) | 84.65 (0.38) | 90.40 (0.24) | 92.66 (0.36) | 88.88 (0.51) | 95.34 (0.48) | 97.56 (0.43) |

Table 10: Table 1 results along with 95 % confidence interval shown in parentheses.

| Method | ChestX | | | ISIC | | | EuroSAT | | | CropDisease | | |
|---|---|---|---|---|---|---|---|---|---|---|---|---|
| | 5W5S | 5W20S | 5W50S | 5W5S | 5W20S | 5W50S | 5W5S | 5W20S | 5W50S | 5W5S | 5W20S | 5W50S |
| MatchNet | 22.40 (0.7) | 23.61 (0.86) | 22.12 (0.88) | 36.74 (0.53) | 45.72 (0.53) | 54.58 (0.65) | 64.45 (0.63) | 77.10 (0.57) | 54.44 (0.67) | 66.39 (0.78) | 76.38 (0.67) | 58.53 (0.73) |
| MatchNet+FWT | 21.26 (0.31) | 23.23 (0.37) | 23.01 (0.34) | 30.40 (0.48) | 32.01 (0.48) | 33.17 (0.43) | 56.04 (0.65) | 63.38 (0.69) | 62.75 (0.76) | 62.74 (0.90) | 74.90 (0.71) | 75.68 (0.78) |
| MAML | 23.48 (0.96) | 27.53 (0.43) | – | 40.13 (0.58) | 52.36 (0.57) | – | 71.70 (0.72) | 81.95 (0.55) | – | 78.05 (0.68) | 89.75 (0.42) | – |
| ProtoNet | 24.05 (1.01) | 28.21 (1.15) | 29.32 (1.12) | 39.57 (0.57) | 49.50 (0.55) | 51.99 (0.52) | 73.29 (0.71) | 82.27 (0.57) | 80.48 (0.57) | 79.72 (0.67) | 88.15 (0.51) | 90.81 (0.43) |
| ProtoNet+FWT | 23.77 (0.42) | 26.87 (0.43) | 30.12 (0.46) | 38.87 (0.52) | 43.78 (0.47) | 49.84 (0.51) | 67.34 (0.76) | 75.74 (0.70) | 78.64 (0.57) | 72.72 (0.70) | 85.82 (0.51) | 87.17 (0.50) |
| RelationNet | 22.96 (0.88) | 26.63 (0.92) | 28.45 (1.20) | 39.41 (0.58) | 41.77 (0.49) | 49.32 (0.51) | 61.31 (0.72) | 74.43 (0.66) | 74.91 (0.58) | 68.99 (0.75) | 80.45 (0.64) | 85.08 (0.53) |
| RelationNet+FWT | 22.74 (0.40) | 26.75 (0.41) | 27.56 (0.40) | 35.54 (0.55) | 43.31 (0.51) | 46.38 (0.53) | 61.16 (0.70) | 69.40 (0.64) | 73.84 (0.60) | 64.91 (0.79) | 78.43 (0.59) | 81.14 (0.56) |
| MetaOpt | 22.53 (0.91) | 25.53 (1.02) | 29.35 (0.99) | 36.28 (0.50) | 49.42 (0.60) | 54.80 (0.54) | 64.44 (0.73) | 79.19 (0.62) | 83.62 (0.58) | 68.41 (0.73) | 82.89 (0.54) | 91.76 (0.38) |
| STARTUP | 26.94 (0.94) | 33.19 (0.46) | 36.91 (0.50) | 47.22 (0.61) | 58.63 (0.58) | 64.16 (0.58) | 82.29 (0.60) | 89.26 (0.43) | 91.99 (0.36) | **93.02** (0.45) | **97.51** (0.21) | **98.45** (0.17) |
| CHEF | 24.72 (0.14) | 29.71 (0.27) | 31.25 (0.20) | 41.26 (0.34) | 54.30 (0.34) | 60.86 (0.18) | 74.15 (0.27) | 83.31 (0.14) | 86.55 (0.15) | 86.87 (0.27) | 94.78 (0.12) | 96.77 (0.88) |
| FT-All | 25.97 (0.41) | 31.32 (0.45) | 35.49 (0.45) | 48.11 (0.64) | 59.31 (0.48) | **66.48** (0.56) | 79.08 (0.61) | 87.64 (0.47) | 90.89 (0.36) | 89.25 (0.51) | 95.51 (0.31) | 97.68 (0.21) |
| ATA | 24.43 (0.2) | – (–) | – (–) | 45.83 (0.3) | – (–) | – (–) | 83.75 (0.4) | – (–) | – (–) | 90.59 (0.3) | – (–) | – (–) |
| ConFeSS | **27.09** (0.24) | **33.57** (0.31) | **39.02** (0.12) | **48.85** (0.29) | **60.10** (0.33) | 65.34 (0.45) | **84.65** (0.38) | **90.40** (0.24) | **92.66** (0.36) | 88.88 (0.51) | 95.34 (0.48) | 97.56 (0.43) |

## J  FEATURE MASKING AND VC THEORY

The generalization ability of a machine learning model is related to the Vapnik-Chervonenkis (VC) theory. The VC dimension (Shawe-Taylor & Cristianini, 2000) measures the capacity or complexity of a machine learning model. For a model family, the VC dimension is the maximum number of training points that can be shattered by that family. The VC dimension of a set of separating hyperplanes is $d + 1$ where $d$ is the feature space dimensionality. Vapnik proved that with probability $1 - \eta$, the test loss ($\mathcal{L}_{te}$) is upper bounded as

$$\mathcal{L}_{te} \leq \mathcal{L}_{tr} + \sqrt{\frac{\gamma + \log(2N) - \log(\frac{\eta}{4})}{N}}, \tag{10}$$

where $\mathcal{L}_{tr}$ is the training loss, $N$ is the number of training samples, and $\gamma$ is the VC dimension. For better generalization, the goal is to reduce the upper bound, which can be decreased by having more training samples $N$. However, when $N$ is small in the few-shot setting, the upper bound increases, triggering generalization performance to drop. If we reduce $\gamma$, we can decrease the upper bound. For a linear classifier, $\gamma$ is upper bounded by the number of features. Hence, if we reduce the number of features, we also reduce the upper bound of $\gamma$ and subsequently the generalization upper bound. This is realized with the masking module $M(\cdot)$, which selects a fraction of features before forwarding them

Table 11: Table 2 results along with 95 % confidence interval shown in parentheses.

| Setting | 5-way 5-shot | | | | 5-way 20-shot | | | |
|---|---|---|---|---|---|---|---|---|
| | CropDisease | EuroSAT | ISIC | ChestX | CropDisease | EuroSAT | ISIC | ChestX |
| Full Framework | 88.88 (0.51) | 84.65 (0.38) | 48.85 (0.29) | 27.09 (0.24) | 95.34 (0.48) | 90.40 (0.24) | 60.10 (0.33) | 33.57 (0.31) |
| w/o Cont. Learn. | 87.26 (0.32) | 83.15 (0.21) | 47.66 (0.18) | 26.06 (0.42) | 95.47 (0.53) ↑ | 88.78 (0.17) | 59.96 (0.28) | 32.12 (0.28) |
| w/o FT BB | 85.18 (0.42) | 83.14 (0.25) | 42.25 (0.73) | 25.76 (0.52) | 93.52 (0.56) | 89.70 (0.25) | 52.61 (0.34) | 31.12 (0.24) |
| w/o Feature Mask | 87.57 (0.44) | 83.87 (0.26) | 47.10 (0.13) | 26.09 (0.15) | 95.49 (0.26) ↑ | 89.93 (0.20) | 61.08 (0.24) ↑ | 33.20 (0.18) |
| w/o $L_{div}$ | 87.95 (0.24) | 84.23 (0.21) | 48.62 (0.22) | 26.92 (0.50) | 94.74 (0.34) | 89.31 (0.25) | 59.20 (0.25) | 32.77 (0.09) |
| w/o $L_{neg}$ | 89.03 (0.12) ↑ | 84.41 (0.36) | 48.04 (0.18) | 26.60 (0.30) | 94.72 (0.17) | 90.37 (0.38) | 60.14 (0.18) ↑ | 32.81 (0.34) |
| w/o $L_{reg}$ | 87.83 (0.30) | 83.98 (0.26) | 48.34 (0.32) | 26.73 (0.24) | 94.43 (0.22) | 90.31 (0.22) | 59.83 (0.08) | 32.69 (0.12) |
| Direct Positive | 87.15 (0.16) | 83.94 (0.15) | 47.25 (0.24) | 26.58 (0.14) | 93.65 (0.21) | 89.40 (0.11) | 59.66 (0.16) | 31.92 (0.20) |

to the linear classifier. Thus, feature selection has theoretical support for improving generalization performance in the few-shot setting. Also, empirical results in Fig. 2 (a) show different datasets and shots selecting different number of features and hence realizing different upper bounds of $\gamma$.

## K  THEORY OF CONTRASTIVE LEARNING AND FEW-SHOT LEARNING

In (Cao et al., 2021), the authors proved the following bound:

$$\mathcal{L}_{sup} \leq \gamma_0 \mathcal{L}_U^- + \gamma_1 s(f_k). \tag{11}$$

Here, $\mathcal{L}_{sup}$ is the supervised evaluation metric for learned representations. $\mathcal{L}_U^-$ is the unsupervised contrastive evaluation metric for true negative samples. $s(f_k)$ is the intra-class deviation using the key encoder $f_k$. $\gamma_0$ and $\gamma_1$ are coefficients depending on class distributions. $\mathcal{L}_{sup}$ can be the training loss of any supervised few-shot meta-learning method which can generalize to novel categories. Since $\mathcal{L}_U^-$ upper bounds $\mathcal{L}_{sup}$, decreasing $\mathcal{L}_U^-$ amounts to decreasing $\mathcal{L}_{sup}$. Also, $\mathcal{L}_U^-$ can be decreased arbitrarily because it is evaluated only on true negative samples. Hence, contrastive losses can be useful for learning representations that are effective for few-shot learning.

## L  THEORY OF DISTANCE-BASED REGULARIZATION FOR FINE-TUNING

In (Gouk et al., 2021), the authors proved that with probability $1 - \eta$, the test loss ($\mathcal{L}_{te}$) is upper bounded as

$$\mathcal{L}_{te} \leq \mathcal{L}_{tr} + \kappa \sum_{j=1}^{L} \frac{D_j^F}{2B_j^F \prod_{i=1}^{j} \sqrt{n_i}} \prod_{j=1}^{L} 2B_j^F \sqrt{n_j} + 3\sqrt{\frac{\log(2/\eta)}{2m}}. \tag{12}$$

Here, $\mathcal{L}_{tr}$ is the training loss. $\kappa$ is a coefficient depending on the properties of dataset. $m$ is the number of training samples. $B_j^F$ is the upper bound of the Frobenius norm of weight parameter of layer $j$ of both pre-trained and fine-tuned model. $D_j^F$ is the upper bound of the Frobenius norm of the difference between weight parameter of layer $j$ of pre-trained and fine-tuned model. $n_j$ is the number of columns in weight parameter of layer $j$. According to the bound, the generalization gap between $\mathcal{L}_{tr}$ and $\mathcal{L}_{te}$ decreases if $D_j^F$'s of all the layers can be decreased. However, minimizing the weights between pre-trained and fine-tuned model for all layers might be cumbersome. Hence, we choose to minimize the Frobenius norm of difference in features for our regularization term $L_{reg}$.

## M  LIMITATIONS OF OUR FRAMEWORK

Although our framework produces competitive performance on the CDFSL benchmark, it has the following limitations: (a) In online setting, when target domain samples arrive in a streaming

fashion, our method might not be applicable. This is mainly because of the presence of the mask generator. Even though the mask generator is a small network, it still requires a small batch of samples for learning the parameters. In the online setting, samples arrive one at a time, and the small masking network might overfit. A workaround to prevent overfitting can be selectively updating only certain parameters during online learning. (b) Another limitation of our framework is the use of large number of hyperparameters in the adaptation step for weighing the loss functions i.e. $\lambda_{pos}$, $\lambda_{neg}$, $\lambda_{div}$ and $\lambda_{reg}$. For practical few-shot adaptation, it is difficult to set aside sufficient number of validation samples to tune the optimal hyperparameter configuration. Hence, we just fixed the hyperparameter values in our experiments. Another possible workaround involves learning hyperparameters themselves within the framework of multi-task uncertainty (Kendall et al., 2018).

