# OpenReview forum: "ConFeSS: A Framework for Single Source Cross-Domain Few-Shot Learning"
_ICLR.cc/2022/Conference — ICLR 2022 Poster_

### Official Review · Reviewer_nRRt · 2021-10-31

**Correctness:** 4
**Technical Novelty And Significance:** 2
**Empirical Novelty And Significance:** 3
**Recommendation:** 6
**Confidence:** 4

**Main Review:**

The challenge is to surpass those approaches which deal on FSL with meta learning. My main claim is: why do not compare with standard approaches from Multi-task learning, Embedding learning, Learning with External Memory? FSL by Meta Learning is just one way, not necessarily the best one. In the experimental results, some approaches have been considered. They should be mapped with respect to a wider taxonomy of FSL. To help the authors, in the introduction ("Few-shot learning methods aim to uncover…"), I suggest to add general references about FSL, such as WANG, Yaqing, et al. Generalizing from a few examples: A survey on few-shot learning. ACM Computing Surveys (CSUR), 2020, 53.3: 1-34. This to justify the meta train-meta test cycle, and to make clear that FSL is not just meta learning.

Why do we need to stick with a single source domain, when there are plenty of universal representation to start the FSL, as the same authors discuss, with (Dvornik et al., 2020)?

Why do we need to apply unsupervised training on the single source domain, instead of a supervised one? Single source domain should be known, hence labeled training data should be available to train with supervision. Perhaps this has to create a general vocabulary of features, due to the distance between domain and target, but this should be better explained.

Are the authors sure that the framework can be cast as meta learning? In general, in meta learning there should be an outer loop and an inner loop, where the outer loop modifies the parameters for the inner one. Instead of this nested general algorithm, the present one is more a serial 3-step architecture, where each step is done just one time. I agree that each step has a double loop inside, in fact this is just a question for clarification.

What is strange to me is the selection of so many hyper parameters, 8. I'm ok with the Hyper-parameter sensitivity ablation study, which is useful for the lambdas, but how do they have been found, especially the number of training epochs for the three steps, which appear so different (400, 15, 50)?

In the appendix H, a general reference to introduce the VC dimension should be given (such as the N. Cristianini and J. Shawe-Taylor one)


**Summary Of The Paper:**

The problem is to learn by few shots (FSL) when the final domain is highly distant from the source base (i.e. natural, medical and satellite), dubbed single source cross-domain few-shot learning. In short, the approach consists of three steps: 1) train a feature extracting backbone with the contrastive loss on a (single source) base category; 2) train a masking module to select relevant features for the target domain; 3) fine-tuned along with the backbone to give features similar to the relevant ones.  The last two are claimed to be novel and, together with the experiments on CDFSL benchmark, sold as the main contributions.

**Summary Of The Review:**

The paper is ok to me, even if it not so general, since it cope with a very specific configuration (single-source domain, FSL with meta learning) which could be just one case of a more general situation, which may lead to better results (multi source domain, or domain which are kind of closer to the final ones). Experiments seem convincing.

---

> ### Author Response · Authors · 2021-11-18
> **Response to Reviewer nRRt**
>
> We thank reviewer nRRt for their insightful comments and suggestions to improve the quality of the paper. Please find the responses below as well as our revised paper.
>
> Comment: The challenge is to surpass those approaches which ... FSL is not just meta learning.
>
> Response: We thank the reviewer for suggesting alternative methods of few-shot learning for comparison. We will consider these methods in our future work. We agree with the reviewer that meta-learning is not the only way to carry out FSL. However, in this work, we only consider FSL works that have been evaluated on this challenging CDFSL benchmark (Guo et al., 2020). Coincidentally, those works seem to be mostly meta-learning based. We have also added the reference about the survey of few shot learning in the second paragraph of the introduction.
>
> Comment: Why do we need to stick ... with (Dvornik et al., 2020)?
>
> Response: We use the single source domain setting mainly for the sake of fair comparison with previous works on this benchmark which mostly use single source domains. Single source data is easier to obtain than collecting the data from multiple sources. Also, with multiple source datasets, we may have the benefit from larger and diverse data, but, we need an additional techniques in the pre-training stage, i.e. domain invariant learning or domain generalization. Although we started with the single source domain setting in this study, we may consider the multi-domain multi-source setting to improve our performance further for the future work.
>
> Comment: Why do we need to apply unsupervised training ... should be better explained
>
> Response: Our justification for using unsupervised pre-training instead of supervised pre-training can be explained both empirically and conceptually. In Table 2, we had shown the ablation (i.e. w/o Cont. Learn.) when unsupervised pre-training is replaced by supervised pre-training. Results show that unsupervised pre-training performs better compared to supervised pre-training. Conceptually, supervised learning tends to create features that are highly discriminative for a particular task. This leads to collapsed features  (Doersch et al., 2020) which cannot generalize well to distant domain tasks. On the other hand, contrastively learned networks focus more on mid and low-level features, which are easily adapted (Zhao et al., 2021). These features produce better reconstruction by learning a holistic representation of images rather than focusing only on discriminative regions. These arguments had been given in the fourth paragraph of Section 1, in Section 2 for Self-supervision for Few-shot Learning as well as in Section 3.2. Furthermore, in appendix K, we provide theoretical support for using contrastive learned representations in few-shot learning.
>
> Comment: Are the authors sure ... question for clarification.
>
> Response: In contrast to the meta-learning frameworks (e.g., MAML) which have inner and outer loops, our method uses a 3-step serial approach, as rightly mentioned by the reviewer. The 3 steps consist of pre-training with contrastive loss, learning of masking module, and fine-tuning. The contrastive training does have two loops - one over samples in the batch and the other one over augmentations of each sample, which is different from the outer loop and inner loop combinations of meta-learning frameworks.
>
> Comment: What is strange to me ... different (400, 15, 50)?
>
> Response: In few-shot learning, since the amount of training data is so less, it is difficult to set aside a validation set to find the optimal hyper-parameters automatically. Hence, the hyper-parameters have been found manually and fixed. Hyper-parameter sensitivity studies for lambdas are shown in Fig. $3$, while those for epochs are shown in Table $8$. The epoch number of 400 for pre-training is chosen according to (Guo et al., 2020), which had been mentioned in Section $4.2$.
>
> Furthermore, there are differences in epoch numbers (400, 15, 50) for different stages of training. Pre-training has higher number of epochs (400) because the feature extraction network is trained with abundant data. On the other hand, the epoch number for training masking module (15) and fine-tuning (50) are rather small to prevent over-fitting on few-shot data. This argument has also been added in appendix G.
>
> Comment: In the appendix H ... such as the N. Cristianini and J. Shawe-Taylor one
>
> Response: We have introduced the reference in the first paragraph of this appendix section (now changed to appendix J).

---

### Official Review · Reviewer_yTZ7 · 2021-11-02

**Correctness:** 4
**Technical Novelty And Significance:** 3
**Empirical Novelty And Significance:** 2
**Recommendation:** 6
**Confidence:** 5

**Main Review:**

Paper strengths:
- The cross-domain few-shot learning problem studied in this paper is a fundamental task that deserves further study.
- The paper is well written and easy to follow.
- The proposed method is straightforward.
- Experiments are sufficient, and ablation studies can bring good observations and analyses for the method.

Paper weaknesses:
- The proposed method consists of three stages for training, which seems complicated and a little bit tricky in implementations. Could it be simplified in the training process?
- In experiments, most results of the proposed method are satisfactory. However, on CropDisease, it is significantly worse than the results of STARTUP. Are there any reasons for the observations?


**Summary Of The Paper:**

This paper proposes a framework, terms as ConFeSS, for dealing with cross-domain few-shot learning problems. Specifically, it firstly trains a feature extracting backbone with the contrastive loss on the base category data for learning better features. Then, it trains a masking module to select relevant features suited to target domain classification. Finally, a classifier is fine-tuned along with the backbone such that the backbone produces features similar to the relevant ones. Experiments are conducted on several cross-domain few-shot learning benchmarks.

**Summary Of The Review:**

The authors are encouraged to respond to the comments listed in the paper weaknesses during rebuttals.

---

> ### Author Response · Authors · 2021-11-18
> **Response to Reviewer yTZ7**
>
> We thank reviewer yTZ7 for their insightful comments and suggestions to improve the quality of the paper. Please find the responses below as well as our revised paper.
>
> Comment: The proposed method consists ... simplified in the training process?
>
> Response: Currently, our training consists of three steps: pre-training the source feature extractor, training the masking module and fine-tuning the target feature extractor. Alternatively, we could combine the second and third step i.e. jointly train the masking module and target feature extractor. Experiments using the joint training method is shown in appendix H in Table 9 with row marked as "Joint Training". Joint training of masking module and target feature extractor produces poorer recognition performance for 5 shot setting compared to ConFeSS. However, for higher shot setting, the joint training procedure produces similar or better performance compared to ConFeSS. This is because joint training encompasses optimization of larger number of parameters, which might cause the network to overfit on lower shots while exploit additional amount of training data for higher shots.
>
> Comment: In experiments, most results ... any reasons for the observations?
>
> Response: It is to be noted that STARTUP uses large amount of unlabelled target domain dataset along with the source dataset in the pre-training stage. Also, among the four target datasets, CropDisease is the most similar to the source dataset miniImageNet. Since the target domain data is used in an unsupervised way and is very similar to the source dataset, STARTUP may somehow leverage these samples in the pre-training stage in a better way. Hence, STARTUP performs better on CropDisease compared to our method and other baselines which only use few shot labelled data in the target domain.

---

### Official Review · Reviewer_MRXL · 2021-11-03

**Correctness:** 3
**Technical Novelty And Significance:** 3
**Empirical Novelty And Significance:** 3
**Recommendation:** 6
**Confidence:** 4

**Main Review:**

Strengths:

1. The paper is well-written and the easy to read. The organization is also clear.
2. The idea is interesting and makes sense. The authors have clear motivation on each step of the proposed framework. Pretrain the network unsupervisedly is natural in the case of cross-domain few-shot learning due to the domain difference between the source domain and target domain. The feature selection module is also designed with a clear motivation for cross-domain few-shot learning. The decoupling of positive features and negative features is shown to be helpful.
3. The authors also conduct extensive analysis of the proposed method.


Weaknesses:

1. The proposed framework has a lot of hyperparameters to tune which is an issue for few-shot learning since there is no validation set to use.

2. No standard deviations are reported in the tables which is important for the evaluation of the few-shot learning due to the randomness.


More questions:

1.  How about using negative features for fine-tuning? This can be a good ablation to show the usefulness of positive features.

2. Another ablation is to pre-train supervisedly to show the advantages of self-supervised pretraining.

3. How the mask module is trained? And would a better mask module produces better final performance?


**Summary Of The Paper:**

In this paper, the authors propose a generic framework for cross-domain few-shot learning. There are three steps: 1. Pre-training the backbone unsupervisedly using a self-supervised contrastive loss. 2. Select relevant features via a mask module. 3. Fine-tune the network with the selected features. The proposed framework is evaluated in the recently proposed CDFSL benchmark. The results show that the proposed framework outperforms the baselines in most of the cases.

**Summary Of The Review:**

In this paper, the authors propose a framework for cross-domain few-shot learning. The idea is reasonable and the authors also provide a good analysis of the components of the framework. The proposed framework outperforms the baselines in most of the cases.

---

> ### Author Response · Authors · 2021-11-18
> **Response to Reviewer MRXL**
>
> We thank reviewer MRXL for their insightful comments and suggestions to improve the quality of the paper. Please find the responses below as well as our revised paper.
>
> Comment: The proposed framework ... there is no validation set to use.
>
> Response: We agree with the reviewer that in few-shot learning, since the amount of training data is so less, it is difficult to set aside a validation set to find the optimal hyper-parameters automatically. Hence, the hyper-parameters have been found manually and fixed. Additionally, hyper-parameter sensitivity studies for lambdas are shown in Fig. $3$, while those for epochs are shown in Table $8$.
>
> Comment: No standard deviations ... due to the randomness.
>
> Response: In appendix I, we re-report comparison studies: Table 10 and Table 11 show the performance with 95 \% confidence interval for Table 1 and Table 2, respectively. We did not add this in the main paper due to space and page limit constraints.
>
> Comment: How about using negative ... of positive features.
>
> Response: We have added this experiment in appendix H. Specifically, we consider two variations:  (a) Neg. Reg.: In this setup, we use negative features for the regularization. Specifically, we use $|| f_i^t - f_i^- ||^2_2$ in Eq. 8 instead of $|| f_i^t - f_i^+ ||^2_2$. (b) Dir. Neg.: We also consider the Direct Negative setting for the ablation study, where instead of using $L_{reg}$, we directly use the negative features obtained using the mask generator to fine-tune the feature extractor and classifier. The results are shown in Table 9. Results show that both Neg. Reg. and Dir. Neg. perform poorly compared to ConFeSS, suggesting that positive features are essential for fine-tuning.
>
> Comment: Another ablation ... of self-supervised pretraining.
>
> Response:  In Table 2, we had shown this ablation (i.e. w/o Cont. Learn.) when self-supervised pre-training is replaced by supervised pre-training. Results show that self-supervised pre-training performs better compared to supervised pre-training. This is because self-supervision alleviates the supervision collapse problem as explained in the introduction (Section 1).
>
> Comment: How the mask module ... better final performance?
>
> Response: The masking module is trained with few-shot samples on top of the pre-trained source feature extractor while keeping the parameters of the source feature extractor frozen. The loss function used for training is $L_{mask}$ in Eq.(6), the details of which are described in Section 3.3. Currently, for the masking module, we use a two layer ReLU based network with a hidden dimension of 256. To study whether better masking module helps, we tried increasing the size of the masking module. The results are shown in appendix H for 3, 4 and 5 - layered masking module in Table 9 with rows marked as 3 Layer Mask Mod., 4 Layer Mask Mod. and 5 Layer Mask Mod., respectively. In most of the cases, having a larger masking module produces similar or slightly better performance compared to ConFeSS because of better representation capacity of output masks.

---

### Official Review · Reviewer_H3UB · 2021-11-08

**Correctness:** 3
**Technical Novelty And Significance:** 2
**Empirical Novelty And Significance:** 2
**Recommendation:** 5
**Confidence:** 4

**Main Review:**

## Strengths
1. The positive and negative features separation is novel, which serves as a regularization for fine-tuning on the target domain.
2. The algorithm 1 is clean and easy to understand.
3. The performance on CDFSL outperforms several SOTA methods.

## Weakness
1. The feature masking module has too many losses making it difficult to reproduce. Comparing to fine-tuning full network, the improvements are small.
2. The ablation study presented in Sec 4.4 is either inefficient or unclear. For example, can we replace eq. (8) with || f_i^t - f_i^+ ||^2?
3. Clarity can be improved:
a) The labels on D_s are discarded?
b) How do you prevent trivial masks? I.e., all close to 1's or all close to 0's?
c) The intuition on the losses in eq.(6) and (8) are not very straightforward. It would be better to start from the simplest case without any of these losses and then add them one by one. How did you remove the mask module in Table 2?
d) Why do you train the mask generator on D_t rather than D_s?
e) How do you tune hyper-params if no validation set on target domain doesn't exist.

**Summary Of The Paper:**

This paper proposes a framework for cross-domain few-shot learning. The framework consists of three steps: 1) pre-training backbone; 2) meta-learn the feature masking network; 3) fine-tuning on target domain. Experiments on the CDFSL dataset show this framework outperforms SOTA methods.

**Summary Of The Review:**

The method is technically sound, and the paper is easy to follow. However, there are still many details are either hidden or unclear, which hinders its reproducibility. Besides, the method is only evaluated on CDFSL, which is a relatively small dataset. It would be more convincing if the method is applied and compared on Meta-Dataset (another larger cross-domain dataset). Hence, this is a borderline paper to me.

---

> ### Author Response · Authors · 2021-11-18
> **Response to Reviewer H3UB (1/2)**
>
> We thank reviewer H3UB for their insightful comments and suggestions to improve the quality of the paper. Please find the responses below as well as our revised paper.
>
> Comment: The feature masking module ... difficult to reproduce.
>
> Response: The feature masking module as described in Eq. (6) has three loss terms. The values of the weight coefficients (lambdas), training epochs, temperature of gumbel sigmoid are  described in Section 4.2. The details of the MMD loss used for $L_{div}$ in Eq. (6) are given in appendix A and B. We also added the architecture details of the masking module in appendix A. So, the results can be reproduced with these information.
>
> Comment: The ablation study ... with $|| f_i^t - f_i^+ ||^2$?
>
> Response: The writing of the ablation study in Section 4.4 has been improved. We also carried out the ablation study where the L2 norm in Eq. (8) is replaced by the L1 norm. Results are shown in the row "L1 norm" in Table 9 in appendix H. Compared to ConFeSS, regularization with L1 norm produces drop in performance for different shots and datasets. This suggests that regularization with L2 norm is advantageous. Furthermore, in Table 9, we also conducted other ablation studies/alternative model designs as suggested by other reviewers.
>
> Comment: The labels on $D_s$ are discarded?
>
> Response: Yes, we use self-supervised pre-training on the source dataset. In Table 2, we had shown an ablation (i.e. w/o Cont. Learn.) when self-supervised pre-training is replaced by supervised pre-training. Results show that self-supervised pre-training performs better compared to supervised pre-training. This is because self-supervision alleviates the supervision collapse problem as explained in the introduction (Section 1).
>
> Comment: How do you prevent ... close to 0's?
>
> Response: Experimental results in Fig. 2 (a) show that the average number of features selected for different shots and datasets ranges from 260-330. If we had trivial masks, the number of features selected would be close to 0 or 512, which is the original feature dimension. Conceptually, the trivial mask of all close to 0's is prevented through $L_{pos}$ which tries to produce discriminative positive features $\mathbf{f}_i^{+} = \mathbf{m}_i \odot \mathbf{f}_i$. If all elements of $\mathbf{m}_i$ were close to 0, we would not have discriminative $\mathbf{f}_i^{+}$. Another possible reason why trivial masks of both fully 0's and 1's are not produced is due to the gumbel sigmoid operation. The gumbel sigmoid operation adds logistic noise to the logits of the mask (described in section 3.3), hence providing a more stochastic optimization. Thus uniform mask values are rarely obtained.
>
> Comment: The intuition on the losses ... add them one by one.
>
> Response: The reviewer probably refers to using additive way of doing ablation studies in Table 2. Although additive approach could be a way of carrying out ablation, it is sometimes rather difficult to clearly see the contribution of each loss term. For example, to find the contribution of L3 we have to select whether to find difference between L1 and L1 + L3 or L1 + L2 and L1 + L2 + L3. On the other hand, Total loss w/o L3 would give a clear indication that when w/o L3, there is a drop in recognition performance, the loss term L3 is necessary. Hence, we use this elimination-based approach to carry out ablation studies in Table 2.
>
> Comment: How did you remove the mask module in Table 2?
>
> Response: In Table 2, w/o Feature Mask describes the setting when the pre-trained backbone is directly fine-tuned using only cross-entropy loss on the few-shot target dataset, skipping the step of learning a masking module. This description has been briefly mentioned in the Ablation Study of Sec 4.4.
>
> Comment: Why do you train the mask generator on $D_t$ rather than $D_s$?
>
> Response: In the ConFeSS framework, the mask generator is trained on the target dataset because we want to select features that are relevant for the target domain during few-shot adaptation. We also performed experiments where the mask generator is trained on the source dataset and then used for regularized fine-tuning on the target dataset. Results are shown in the row "Source Mask" in Table 9 in appendix H. Compared to ConFeSS, training the masking module on the source dataset produces poorer performance.
>
> Comment: How do you tune hyper-params ... target domain doesn't exist.
>
> Response: The hyper-parameters have been found manually and fixed. Additionally, hyper-parameter sensitivity studies for lambdas are shown in Fig. $3$, while those for epochs are shown in Table $8$. These sensitivity studies give an idea of the ideal operating hyper-parameter configuration.

---

> > ### Author Response · Authors · 2021-11-18
> > **Response to Reviewer H3UB (2/2)**
> >
> > Comment:  It would be more convincing if the method is applied and compared on Meta-Dataset (another larger cross-domain dataset)
> >
> > Response: In this work, we considered the BCDFSL benchmark (Guo et al., 2020) because it was more recently introduced and focuses specifically on the CDFSL problem. Even if the method is only evaluated on CDFSL, we performed a lot of ablation studies to thoroughly evaluate the effectiveness of our method. With this learned knowledge, we will apply our method to another bigger dataset (e.g., Meta-Dataset) for our future work.

---

### Public Comment · ~Haoqing_Wang1 · 2021-11-19
**Missing important baselines**

Hi, [1] is not considered in you paper, which has been accepted by IJCAI 2021. Importantly, they use a simpler method and achieve high accuracy than yours in 5way-5shot tasks on ChestX, ISIC, EuroSAT and CropDisease. I think you should use [1] as the baseline, or explain the reason why it cannot be used as a baseline.

[1] Cross-Domain Few-Shot Classification via Adversarial Task Augmentation. IJCAI 2021.

---

> ### Author Response · Authors · 2021-11-23
> **Response to Haoqing Wang**
>
> Thank you for bringing your published paper [1] to our notice. It is an impressive work that produces good performance on the BCDFSL benchmark (Guo et al.). We have read your paper and can argue that the method of adversarial task augmentation (ATA) is a plug and play method that needs to be added on top of existing methods to boost their performance. On the other hand, our paper takes a complementary direction and proposes a stand alone and holistic framework that can still achieve competitive performance on this BCDFSL benchmark.
>
> In [1], two tables - Table 1 and Table 2 are used to report comparison studies. Compared to the best obtained results of ATA in Table 1 in [1] for 5-way/5-shot, our method shows better results for 3 out of 4 target datasets. ATA performs slightly better only on the CropDisease dataset, which has the least domain difference with the source dataset miniImageNet.
>
> The Table 2 in [1] shows the results when the support set is augmented with hand-crafted methods to have additional 15 samples [2] (e.g., gamma correction, flip and rotation).
> Compared to the best results of ATA in this table, our method can still perfom better on ChestX dataset, which has the largest domain diffference with the source dataset miniImageNet.
>
> As for the comparison studies of Table 1 in our paper, we don’t consider the augmented support set proposed in Table 2 of [1], as we we feel it does not allow for fair and direct comparison. However, we will consider it for future work since we think that our method would be also improved. In fact, Table 1 of our paper shows that the performance of our method improved rapidly with increasing number of support samples (20 and 50 shots). For the 20 shot setting of our method compared to 5 + 15 shot setting of ATA in Table 2 of [1], our method still performs better on 3 out 4 target datasets while producing similar performance on the CropDisease dataset.
>
> Given that ATA [1] is a strong baseline for CDFSL, we will discuss about the method and include the comparison studies in the final version of our paper.
>
> References -
>
> [1] Wang et al., Cross-Domain Few-Shot Classification via Adversarial Task Augmentation. IJCAI 2021
>
> [2] Jia-Fong Yeh et al, "Large margin mechanism and pseudo query set on cross-domain few-shot learning", arXiv:2005.09218, 2020

---

### Public Comment · ~Yi_Rong3 · 2024-06-22
**Training Details**

Could you please provide the number of target samples used for mask learning and fine tune?
I think this is important for readers to fairly compare the proposed method with other benckmark approaches

---

### Decision · Program_Chairs · 2022-01-20

**Decision:**

Accept (Poster)

**Comment:**

Summary:
Paper addresses the cross-domain few-shot learning scenario, where meta-learning data is unavailable, and approaches are evaluated directly on novel settings. Authors propose a 3-step approach: 1) self-supervised pretraining, 2) feature selection, 3) fine-tuning, and demonstrate gains over state-of-art.

Pros:
- Approach is novel for this setting
- Paper is clear and easy to understand
- Performance beats several prior methods
- Experiments are thorough
- Fundamental problem is worthwhile of investigation

Cons:
- Some concerns among multiple reviewers on how hyperparameters are selected. Authors have provided more information and tables in the paper.
- Training process is multi-step and not unified. Authors provided additional information about unified training results, which yielded poorer results, likely due to overfitting from training many parameters at once.

Overall recommendation based on the consensus of reviewers and AC expertise: accept.